# An Update on the Metabolic Landscape of Oncogenic Viruses

**DOI:** 10.3390/cancers14235742

**Published:** 2022-11-23

**Authors:** Ahmed Gaballah, Birke Bartosch

**Affiliations:** 1Pathogenesis of Chronic Hepatitis B and C Laboratory—LabEx DEVweCAN, Inserm U1052, Cancer Research Centre of Lyon, 69424 Lyon, France; 2University of Lyon, 69424 Lyon, France; 3Institut des Sciences Pharmaceutiques et Biologiques, University Lyon 1, 69424 Lyon, France; 4CNRS UMR5286, 69424 Lyon, France; 5Centre Léon Bérard, 69424 Lyon, France; 6Microbiology Department, Medical Research Institute, Alexandria University, Alexandria 5424041, Egypt

**Keywords:** oncogenic viruses, metabolism, viral persistence, cancer

## Abstract

**Simple Summary:**

Cancer cells amplify in an uncontrolled fashion. The resulting tumor and metastases need to ensure their survival in the body. To achieve this, cancer cells display increased nutritional needs and an altered metabolism. These metabolic changes start to be targeted for therapeutic interventions in the context of a number of different cancers. Similar to cancer, cells infected with viruses that cause cancer, so-called “oncoviruses”, have altered nutritional needs to support the amplification and spread of new progeny viruses and to ensure the survival of infected cells in the host. Here, we give an update on the similarities between the metabolic alterations observed in many types of cancers and those induced by oncogenic viruses. Furthermore, we discuss the antiviral activities of metabolic inhibitors used for the treatment of cancer.

**Abstract:**

Viruses play an important role in cancer development as about 12% of cancer types are linked to viral infections. Viruses that induce cellular transformation are known as oncoviruses. Although the mechanisms of viral oncogenesis differ between viruses, all oncogenic viruses share the ability to establish persistent chronic infections with no obvious symptoms for years. During these prolonged infections, oncogenic viruses manipulate cell signaling pathways that control cell cycle progression, apoptosis, inflammation, and metabolism. Importantly, it seems that most oncoviruses depend on these changes for their persistence and amplification. Metabolic changes induced by oncoviruses share many common features with cancer metabolism. Indeed, viruses, like proliferating cancer cells, require increased biosynthetic precursors for virion production, need to balance cellular redox homeostasis, and need to ensure host cell survival in a given tissue microenvironment. Thus, like for cancer cells, viral replication and persistence of infected cells frequently depend on metabolic changes. Here, we draw parallels between metabolic changes observed in cancers or induced by oncoviruses, with a focus on pathways involved in the regulation of glucose, lipid, and amino acids. We describe whether and how oncoviruses depend on metabolic changes, with the perspective of targeting them for antiviral and onco-therapeutic approaches in the context of viral infections.

## 1. Introduction

Reprogramming of energy metabolism is considered as a hallmark of cancer cells. Transformed cells adopt metabolic changes to cope with their increased need for fast growth and proliferation [1]. Glucose is a main source of cellular energy and generally converted via glycolysis in the cytoplasm into pyruvate, which is then transferred into mitochondria, where it serves as substrate for the tricarboxylic acid (TCA) cycle and oxidative phosphorylation (OXPHOS). Under hypoxic conditions, OXPHOS is throttled, and cells mainly rely on glycolysis, which produces ATP in an oxygen-independent manner. However, many cancer cells limit glucose utilization to glycolysis as opposed to OXPHOS even under aerobic conditions. This metabolic anomaly was first observed in the thirties of the last century by the German biochemist Otto Warburg and is known as the Warburg effect or aerobic glycolysis [2,3]. As glycolytic ATP production is faster than OXPHOS, glycolysis copes better with the increased need of energy and accelerated growth and proliferation in cancer cells [4]. Indeed, increased transport and utilization of glucose is frequently observed in cancer cells, which then channel glycolytic intermediates into biosynthetic pathways such as the pentose phosphate shunt (PPP) for nucleotide production and regeneration of the reducing equivalent NADPH, which is important for cellular redox homeostasis and the synthesis of lipids and amino acids. Consistently, metabolic reprogramming of lipid and amino acid metabolism is also frequently described in cancer [5,6,7,8,9,10]. A scheme of the metabolic pathways is depicted in Figure 1.

Viruses play an important role in cancer development and about 12% of cancer types are linked to viral infections [11]. Viruses that induce malignant transformation are known as oncoviruses and include, for example, Epstein–Barr virus (EBV), human papillomavirus (HPV), human T cell lymphotropic virus (HTLV), hepatitis B virus (HBV), hepatitis C virus (HCV), Kaposi’s sarcoma herpesvirus (KSHV), and Merkel cell polyomavirus (MCP) [12]. Although the mechanisms of viral oncogenesis differ, all these oncogenic viruses share the ability to establish persistent chronic infections, often with no obvious symptoms for years, during which they manipulate and alter cell signaling pathways that coordinate cell cycle progression, apoptosis, and inflammation [13]. Furthermore, oncogenic viruses introduce into their host cells metabolic adaptations similar to those observed in cancer cells, as shown in Figure 2, and importantly, it seems that they depend on these changes for their persistence and amplification.

## 2. Reprogramming of Central Carbon Metabolism in Cancer Cells

Metabolic reprogramming is a vital hallmark of malignant cellular transformation. The highly energy-demanding cancer cells often depend on glycolysis as a fast source for energy production [7,14] and for carbon intermediates as substrates for nucleotide, amino acid, and lipid biosynthesis [15,16]. Concomitantly with the increase in glucose uptake, activities of the mitochondrial TCA cycle and OXPHOS are often preserved in cancer [17,18,19,20,21]. These latter pathways ensure not only a sufficient supply of ATP but also the synthesis of metabolic precursors required for increased biomass in proliferating cells, maintenance of cellular redox homeostasis, and adaptation to the tumor microenvironment. In addition to promoting anabolism in cancer cells, the metabolic shift towards glycolysis also favors cell survival under stressful conditions, increases cell invasiveness, and helps cells overcome immune surveillance [22]. Furthermore, amino acid metabolism and, in particular, glutaminolysis are frequently induced in cancer cells to satisfy amongst others the cellular need for nitrogen for biosynthesis of nucleotides, proteins, lipids, and many other molecules. Furthermore, glutamine is important for the regulation of cellular redox homeostasis [23].

### 2.1. Glucose Uptake

The transport of glucose across the cellular plasma membrane is a vital step and precedes glycolysis. Expression of cytoplasmic membrane glucose transporters (GLUTs), responsible for glucose uptake, is frequently deregulated in cancer [24]. GLUT1 expression is upregulated in response to the oncogenes such as rat sarcoma oncogene (RAS), c-mylocytomatosis oncogene (cMyc), and hypoxia-inducible factor 1α (HIF1α) [25,26,27] and inhibited by tumor suppressor p53, phosphatase and tensin homolog (PTEN), and microRNA-22 (mir-22) [28,29,30]. The overexpression of GLUT1 was found to be positively correlated with the disease stage in various cancer types [31,32,33]. Additionally, the use of GLUT1 inhibitor STF-31 induces apoptosis in cancer cells [34,35]. Furthermore, the IKK–NF-κB axis, required for cellular transformation, is activated by the loss of p53, which leads to overexpression of GLUT3 and aerobic glycolysis [36].

### 2.2. Glycolysis

Once across the plasma membrane, glucose is immediately phosphorylated by hexokinases (HKs), which is the first rate-limiting step of glycolysis. Overexpression of HK1 and HK2 in cancer cells has been previously reported, and HK1 overexpression is a poor prognostic marker in colorectal cancer [37,38]. The HK2-mediated Warburg effect is essential for cellular transformation and growth in many cancer types [39]. In prostate cancer cells, loss of PTEN leads to overexpression of HK2 via activation of the Akt/mTORC1/4EBP1 axis and loss of p53 increases the stability of HK2 mRNA by inhibiting the biogenesis of miR-143, which destabilizes HK2 mRNA in cancer cells [39,40]. Importantly, binding of HK1 and 2 to mitochondria also protects cells from apoptosis via retro-translocation of pro-apoptotic Bcl2 family members away from mitochondria, thus inhibiting death receptor-mediated apoptosis [41]. Finally, binding of HK1 to mitochondria reduces glyceraldehyde 3-phosphate dehydrogenase (GAPDH) activity, thus redirecting glycolytic fluxes towards PPP and dampening inflammatory cytokine production [42]. In addition to HK and GAPDH, the expression of many intermediate enzymes of glycolysis, including phosphofrucktokinase (PFK), aldolase (ALODA), phosphoglycerate kinase 1 (PGK1), and alpha enolase (ENO1), is regulated by oncogenes, including cMyc, HIF1α, β-catenin, and tumor suppressor p53. These enzymes are overexpressed in many forms of cancers, which is often associated with poor prognosis and treatment efficacies, and inhibition or knock out of the corresponding genes has been shown to have anti-oncogenic effects [43,44,45,46,47,48,49,50,51].

Pyruvate kinases mediate the final rate-limiting and net ATP-producing step in glycolysis. In cancer cells, the embryonic isoform PKM2 predominates over the adult isoform PKM1 [52]. Switching the expression from PKM2 to PKM1 reverses the Warburg effect, decreases tumorigenicity, and increases oxygen consumption through shunting glycolysis to OXPHOS [53]. In normal cells, the produced pyruvate is converted to acetyl-coA via pyruvate dehydrogenase (PDH), a step that links glycolysis to the TCA cycle and OXPHOS by mediating pyruvate uptake into mitochondria. This reaction is inhibited in cancer cells due to the overexpression of pyruvate dehydrogenase kinases (PDK), a family of PDH complex inhibitors, leading to a reduced acetyl-coA supply to the TCA cycle [54]. It has been shown that knocking down PDK2 in cancer cell lines increases oxygen consumption and reactive oxygen species (ROS) production and induces apoptosis [55]. Noteworthy, HIF1α increases the expression of PDK [54].

In cancer cells, pyruvate produced by aerobic glycolysis is converted to lactate via lactate dehydrogenase (LDH). LDH is a homo- or heterotetramer composed of two subunits, LDHA and LDHB. LDHA expression was found to be upregulated in cancer cells [56]. LDHA inhibitors FX11 and oxamate were found to reduce lactate production and retard the growth of cancer cells [57,58]. Lactate is a key player in the process of carcinogenesis. In many tumors, the level of lactate in cancer cells is elevated up to 40-fold. In cancer cells, the rate of glycolytic fluxes frequently exceeds the capacity of mitochondria to oxidize produced pyruvate due to the limited availability of free CoA molecules and a low NAD^+^/NADH ratio. Overexpression of LDHA offers a faster alternative compared to the TCA cycle to consume pyruvate and regenerate NAD^+^ levels through the production of lactate [6]. Elevated intracellular lactate increases the expression of monocarboxylate transporters MCT1 and MCT4, which mediate the efflux of lactate outside the cells [59]. The release of lactate is sufficient to stimulate angiogenesis and consequently carcinogenesis via activation of vascular endothelial growth factor (VEGF) in tumor endothelial cells [60]. Moreover, high levels of lactate are associated with metastasis in different types of cancer [59]. Lactate supports the immune escape of cancer cells via induction of a state of acidosis in the extracellular microenvironment [61] and by its direct inhibitory effect on immune cells, including macrophages [62] and T cells [63].

### 2.3. Pentose Phosphate Pathway (PPP)

The PPP, a major glucose catabolic pathway, is composed of an oxidative and a non-oxidative phase. The rate-limiting step in the oxidative phase is the dehydrogenation of glucose-6-phosphate (G6P) via glucose-6-phosphate dehydrogenase (G6PD). The products of the oxidative branch of the PPP include not only the nucleoside precursor ribose-5-phosphate (R5P) but also NADPH [64]. The increased metabolic activities in cancer cells are associated with excessive ROS production and increased oxidative stress, which need to be kept in check [65]. In addition to its role as an essential anabolic reducing factor in the biosynthesis of nucleotides and fatty acids (FAs), NADPH is responsible for maintaining cell survival under oxidative stress [66,67]. NADPH is the source of reductive potential used by glutathione reductases (GR) to recycle oxidized glutathione into its reduced form, which is needed for detoxification of ROS [68]. In line with these observations, G6PD activity has been shown to be enhanced in several types of cancer [69,70,71,72]. Several growth factors and oncogenes are capable of increasing the activity of G6PD, including platelet-derived growth factor (PDGF), epidermal growth factor (EGF), phosphoinositide 3-kinase (PI3K), Src, and Ras [73,74,75,76]. On the other hand, the tumor suppressors, p53 and PTEN, reduce the PPP via binding and inhibition of the activity of G6PD [77,78,79].

The reversible non-oxidative phase of the PPP consists of transfer reaction series that result in the production of sugars, including pentoses, for anabolic purposes [80]. This phase of the pathway is under the control of transketolase (TKT) and transaldolase (TALDO), enzymes that link the non-oxidative phase of PPP to glycolysis. TKT catalyzes the reversible conversion of xylulose-5-P and ribose-5-P to glyceraldehyde-3-P and sedoheptulose-7-P and the conversion of xylulose-5-P and erythrose 4-P to glyceraldehyde-3-P and fructose-6-P. TALDO reversibly catalyzes the conversion of glyceraldehyde-3-P and sedoheptulose 7-P to erythrose 4-P and fructose-6-P. Thus, the non-oxidative branch replenishes metabolites of the oxidative branch and regulates the flux of glycolysis by providing fructose 6-P and glyceraldehyde 3-P [81]. Several studies have reported upregulation of TKT and TALDO in several types of cancer [82,83,84]. Moreover, TKT overexpression has been correlated with tumor invasiveness and poor cancer prognosis in lung, prostate, and breast cancer cells [73,85].

### 2.4. Tricarboxylic Acid Cycle (TCA)

The TCA cycle has been recognized as a central hub not only for ATP production but also for the production of biosynthetic precursors. Contrary to the assumption that mitochondrial functions are reduced in cancer cells, many lines of evidence confirm the indispensable activity of the mitochondrial TCA cycle in many different types of cancer. While non-proliferating cells maximize the energy production in the TCA cycle from oxidizable substrates, in cancer cells, the TCA cycle is important for the provision of intermediates for lipid, nucleotide, and amino acid synthesis rather than ATP [6]. This continuous efflux of biosynthetic precursors from the TCA cycle is known as cataplerosis. For cancer cells to preserve TCA cycle activity, cells must provide the necessary substrates. This compensatory supply is known as anaplerosis [86]. Cells use different carbon fuels, including glucose, glutamine, and FA, to maintain anaplerosis [43].

## 3. Reprogramming of Fatty Acid Metabolism

Compared to normal cells, cancer cells are characterized by lipid accumulation in the form of lipid droplets. The lipid requirement is increased during tumorigenesis for the synthesis of membranes and signaling molecules [8]. De novo synthesis of FA is activated in several cancers via the increased expression of the transport protein citrate carrier (CIC) responsible for transporting the mitochondrial citrate to the cytoplasm [87]. Additionally, ATP citrate lyase (ACLY), the first rate-limiting enzyme in FA de novo synthesis, is upregulated in several types of cancer [88,89,90,91]. Overexpression of ACLY is associated with increased tumorigenesis and gene knockout inhibits tumor growth. The cytosolic form of acetyl-coA carboxylase 1 (ACC1) catalyzes another rate-limiting step of FA synthesis, the conversion of acetyl-coA to malonlyl-coA [92]. The tumor suppressor LKB1 inhibits FA synthesis via activation of the central metabolic sensor, adenosine monophosphate activated protein kinase (AMPK), which inactivates ACC1 [93]. The terminal step of de novo FA synthesis is catalyzed by fatty acid synthase (FASN). Higher activity of FASN supports tumor cell survival and overexpression was reported in different types of cancers [94,95].

## 4. Reprogramming of Amino Acid Metabolism

Rapidly proliferating cancer cells have increased requirements for amino acids not only for protein synthesis but also as important metabolites and metabolic regulators. Several amino acids were shown to have special importance in cancer metabolism. In particular, glutamine is considered very important, as the most abundant amino acid. Dependence on glutamine metabolism, also referred to as “glutamine addiction”, is a hallmark of cancer cell metabolism.

Glutamine is converted to α-ketoglutarate via glutaminolysis, which fuels the TCA cycle [96]. Conversion of glutamine into citrate by isocitrate dehydrogenase (IDH) can feed into lipogenesis. Cytoplasmic citrate is converted into malate and then pyruvate in an enzymatic process that produces NADPH required for redox homeostasis [97,98]. Additionally, glutamine can serve as a nitrogen donor to support the increased demand for nucleotide synthesis [96,99,100] and contribute to the biosynthesis of uridine diphosphate-N-acetylglucosamine (UDP-GlcNAc), which is needed for protein glycosylation and endoplasmic reticulum stress responses in cancer cells [101]. Glutamine also plays a pivotal role in regulating the increased levels of ROS in tumor cells as it is a precursor for glutathione synthesis [99].

The expression of cancer-specific isoforms of glutaminase (GLS), a key enzyme of glutaminolysis and responsible for converting glutamine into glutamate, is regulated by the oncogene cMyc [102,103]. cMyc was found to induce glutamine conversion into glutamate under normo- and hypoxic conditions. Under glucose deprivation conditions, glutamine was found to be the main precursor for the synthesis of TCA intermediates such as fumarate, malate, and citrate, confirming the ability of glutamine to fuel the TCA cycle [104]. cMyc also induces the expression of enzymes involved in nucleotide biosynthesis pathways, which increase the need for glutamine as a nitrogen donor for nucleotide biosynthesis [100]. Glutamine metabolism was shown to inhibit autophagy in cancer cells through the activation of the mTOR pathway [105]. The highly selective glutaminase inhibitor CB-839 has been tested in clinical trials for cancer treatment. While monotherapies have failed overall, CB-839 is showing promising effects in some combination therapies [106,107,108,109,110].

Several cancer subtypes hyperactivate the anabolic serine and glycine synthesis side-branch of glycolysis, which provide vital precursors for the synthesis of proteins, nucleic acids, and lipids that are essential to cell proliferation. Additionally, serine/glycine biosynthesis influences the cellular anti-oxidative capacity, thus maintaining tumor homeostasis [111]. The glycolytic intermediate 3-phosphoglycerate (3-PG) is the precursor for de novo serine synthesis. Serine can then be converted into glycine, a process mediated by serine hydroxymethyltransferase 1 or 2 (SHMT1/2). Furthermore, serine contributes to the methionine cycle and impacts the availability of S-adenosyl methionine, the donor of methyl groups for epigenetic regulation of RNA and DNA [112].

Proline is a secondary amino acid that is stored in collagen. Proline metabolism is related to ATP production, protein and nucleotide synthesis, and redox homeostasis. Rewiring of proline metabolism has been shown to be tumor type dependent [113,114,115,116,117]. Proline is also a precursor for polyamine synthesis. Polyamines such as spermidine and spermine are polycationic alkylamines [118]. These small molecules are involved in many central processes of cell survival and proliferation, including protein and nucleic acid biosynthesis, chromatin structure stabilization, differentiation, apoptosis, protection against oxidative stress, and intercellular communication [118,119,120,121]. In cancer cells, polyamine metabolism is often dysregulated, and elevated intracellular polyamine levels are needed for transformation and tumor progression [122].

## 5. Role of Oncogenes, Tumor Suppressors, and Cell Signaling in Regulating Cancer Cell Metabolism

Oncogenes and tumor suppressor genes are key players in the process of carcinogenesis as they regulate cellular proliferation, growth, and/or cell cycle arrest [123]. Many lines of evidence have confirmed and analyzed in depth the roles of these factors in regulating metabolic enzymes and metabolic signaling pathways important for growth, proliferation, and regulation of cellular redox homeostasis [5,124].

### 5.1. PI3K/Akt/mTOR Pathway

The highly conserved PI3K/Akt/mTOR signaling pathway is commonly deregulated in cancer. Activation of this signaling pathway has important roles in tumorigenesis [125]. The binding of growth factors to its receptors activates PI3K, which in turn activates the downstream serine/threonine kinases Akt and mTOR. Activation of this pathway increases the membrane expression of transporters for glucose, amino acids, and other metabolites [126,127,128,129,130]. Activated Akt was found to stimulate the expression and activity of glycolytic enzymes, leading to increased glycolysis and lactate production [131,132,133]. Moreover, PI3K and Akt increase lipid synthesis via an increase in the expression of lipogenic genes [134,135]. On the other hand, mTOR inhibits autophagy, enhances ribosome biogenesis, and increases protein, FA, and nucleotide synthesis [136,137,138,139]. Additionally, mTOR stimulates aerobic glycolysis via an increase in the expression HIF1α transcript levels and its glycolytic gene targets [137].

In normal cells, PI3K signaling is tightly controlled via the lipid kinase PTEN, which counteracts all metabolic features of the PI3K/Akt/mTOR pathway [140]. Systemic overexpression of PTEN in transgenic mouse models resulted in a metabolic phenotype characterized by diminished aerobic glycolysis and glutaminolysis [141]. The tumor suppressor p53 is a key regulator of metabolic homeostasis [142]. p53 is able to inhibit PI3K signaling via activation of the tumor suppressor genes tuberous sclerosis complexes (TSC) 1 and 2, PTEN, and AMPK [143,144].

### 5.2. Hypoxia-Inducible Factor (HIF1α)

Hypoxia, or low oxygen tension, significantly affects the process of tumorigenesis. Under normoxic conditions, HIF1α is hydroxylated, which leads to its immediate degradation via the Von Hippel–Lindau (VHL) E3 ubiquitin ligase complex. Under hypoxic conditions, the implied hydroxylases become inactive, thus leading to HIF1α stabilization [145]. HIF1α is a transcription factor complex that is activated in many cancers and its expression is usually associated with a poor prognosis [146]. It regulates and coordinates cellular responses to hypoxia by inducing genes encoding glucose transporters, glycolytic enzymes, and LDHA in order to enhance glycolysis and lactate production while blocking the access of glycolytic products to mitochondria by targeting PDK1 [147,148,149]. Moreover, HIF1α participates in maintaining a functional TCA cycle via glutamine-driven anaplerosis to compensate the limited supply of glycolytic carbon [21]. Furthermore, it activates the reductive carboxylation of glutamine to provide another carbon source for de novo FA synthesis in vitro [97,150].

### 5.3. cMyc

cMyc, the human homolog of a retroviral gene, is commonly amplified in cancer and plays significant roles in the metabolic adaptation of cancer cells [6]. It can be induced directly or indirectly in response to hormone- or growth-factor-sensitive signal transduction pathways such as PI3K/Akt/mTOR, Wnt/β-catenin, and EGFR/RAS/MAPK [151]. cMyc induces the expression of glycolytic genes, including GLUT1, HK2, and LDHA [27,152]. It promotes the production of the cancer-specific isoform PKM2 by directing the splicing of the pyruvate kinase M pre-mRNA [153]. cMyc is also known to activate glutaminolysis by regulating the expression of GLS isoforms [103,104,154,155] and activate the transcription of enzymes required for nucleotide synthesis [156,157,158]. Finally, cMyc was found to increase the expression of enzymes involved in serine biosynthesis [159]. However, the proliferation of cancer cells was found to mainly depend on the exogenous serine supply via extracellular uptake, with extracellular glycine being unable to compensate for the loss of serine [160]. Intracellular serine may be converted to glycine under the activity of SHMT, which is a direct target for the oncogene cMyc [157].

## 6. Metabolic Reprogramming in Oncoviruses

Many similarities have emerged between the metabolic pathways that are modulated in cancer cells versus those altered in cells infected with oncoviruses, as schematically shown in Figure 2. Furthermore, many of the transcription factors, oncogenes, and tumor suppressors that drive metabolic changes in cancer cells are also targeted by oncoviruses, as shown in Figure 3. While virally induced metabolic changes are generally not considered to drive cellular transformation per se, they may facilitate the transformation process in a setting of oxidative stress, genomic instability, and inflammation, as frequently induced by oncoviruses. Importantly, the replication and spread of many oncogenic viruses depend on the metabolic changes in the host cell, which offers important opportunities to target viral replication and the associated cancer using the same therapeutic targets.

### 6.1. Epstein–Barr Virus (EBV)

EBV is a double helix gamma herpes virus that preferentially infects B lymphocytes and epithelial cells [161]. It has been associated with several malignancies [162,163]. After primary infection in B cells, EBV usually starts a latent infection where latent viral proteins stimulate viral DNA replication and cell proliferation. Alternatively, EBV produces infectious virions during the lytic replication phase [164]. The stimulation of lytic reactivation is induced by transcription transactivator proteins ZEBRA and BRLF1 [165]. Several viral genes with oncogenic activity have been identified in the EBV genome, including six nuclear antigens (EBNA1, EBNA2, EBNA3A, EBNA3B, EBNAC, and EBNA-LP) and two latent membrane proteins (LMP1 and LMP2) [166,167].

Early EBV infection induces a transient hyperproliferation period that is suppressed by DNA damage responses and cell cycle arrest. Proliferation-arrested EBV-infected cells showed decreased expression of TCA cycle and OXPHOS enzymes and reduced mitochondrial respiration compared to uninfected B cells. On the other hand, hyperproliferative EBV-infected cells that transform into lymphoblastoid cells showed increases in both glycolysis and OXPHOS [168]. LMP1 overexpression in nasopharyngeal epithelial cells is associated with increased glycolysis and lactate production via increased expression of HK2, PKM2, and LDHA1 [169,170,171]. In concordance with these finding, LMP1 was found to increase HK2 and GLUT1 expression indirectly through decreasing the expression of homeobox gene C8 or, alternatively, by enhancing the degradation of prolyl HIF-hydroxylases, thus stabilizing HIF1α [172]. EBV infection of B lymphocytes directly promoted temporal induction of MCT1 and MCT4 through the viral proteins EBNA2 and LMP1, respectively [173]. Targeting of a recently identified HK isoform, HK domain component 1, has shown therapeutic effects against EBV-induced lymphoma by modulating mitochondrial functions and suppressing EBV replication in mouse models [174]. Glut1-specific chemical inhibitors have been shown to kill NPC cells in vitro [175]. In addition, EBV-encoded microRNAs are highly expressed in EBV-positive tumors and impact glucose metabolism by targeting PTEN/PI3K/Akt, cMyc, and NF-κB and ERK pathways [176,177].

Additionally, EBV infection also affects amino acid metabolism. In addition to enhancing glycolytic enzymes, LMP1 was found to increase glutamine uptake in nasopharyngeal carcinoma cells [178]. Furthermore, an upregulation of GLS1 isoforms was recently shown to stimulate mitochondrial metabolism and cell proliferation in EBV-infected cells [178]. Cell proliferation and the viability of latently EBV-infected cells were sensitive to pharmacological inhibition of GLS1 isoforms [178]. EBV infection was also found to upregulate extracellular serine uptake and de novo serine synthesis [179]. However, methionine restriction, or methionine cycle perturbation, was shown to cause hypomethylation of EBV genomes and de-repress latent membrane protein and lytic gene expression in Burkitt cells. Additionally, methionine metabolism was shown to regulate EBV latency genes required for B cell immortalization [180]. Finally, mTOR inhibitors reduced the capacity of EBV-positive cells to undergo lytic replication in a cell-type-dependent fashion [181].

With respect to nucleotide metabolism, EBV infection of primary B cells was found to upregulate cytidine triphosphate synthases CTPS1 and CTPS2, which is consistent with the observation that purine dNTP biosynthesis is critical in the early stages of EBV-mediated B cell immortalization [182]. Furthermore, EBNA2 and/or EBNA-LP were needed for the induction of CTPS1 in newly infected B cells. Finally, CTPS1 depletion impaired EBV lytic DNA synthesis [183].

The transcriptional factor, EBNA2, was found to induce cMyc expression [184]. EBNA2 and EBNA5 inhibit ubiquitination-induced degradation of HIF1α [185]. The p53 pathway has been found to be inactivated by EBNA3C [186].

EBV also modulates the host cell lipid metabolism. The EBV viral protein BRLF1 expression induces the expression of FASN via activation of the MAPK pathway [187]. Newly EBV-infected B cells are characterized by increased expression of proteins involved in FA and cholesterol metabolism [188]. In EBV-driven cancer cell lines, inhibition of FASN was associated with a decrease in the BRLF1-mediated lytic viral genes, indicating that FA synthesis is required for EBV viral gene expression [187].

### 6.2. Human Papilloma Virus (HPV)

HPV is a non-enveloped double-stranded DNA virus. Sexually active adolescents are highly susceptible to this infection. The viral genome is composed of three regions: the early region (E), which encodes E1 to E7 proteins necessary for viral replication and cellular transformation; the late region (L), which encodes capsid proteins L1 and L2; and the long control region, which contains the origin of replication and transcription factor binding sites [189]. Low-risk HPV types are associated with formation of benign warts. High-risk HPV types, and in particular HPV 16 and 18, are considered to be the most powerful oncogenic viruses, responsible for about 5% of all cancer cases [190,191,192]. E6 and E7 from high-risk HPV are considered oncoproteins.

The E7 protein of HPV 16 binds to PKM2 and induces a shift from its active tetrameric form to a less active dimeric form in HPV16 E7-transformed 3T3 fibroblasts. The dimeric form is more common in tumors and displays lowered substrate activity [193,194]. High levels of the dimeric form of PKM2 are associated with increased glycolytic intermediates such as fructose biphosphate, essential for amino acid biosynthesis, and increased NADPH biosynthesis, which is needed to regulate ROS levels [195]. To maintain the functionality of the TCA cycle, E7 was shown to promote anaplerotic pathways by activating glutaminolysis [193].

Moreover, E7 degrades the tumor suppressor RB [196] and both E6 and E7 inhibit p53 [197,198,199] and activate the PI3K/Akt/mTOR signaling pathway [200]. HPV16 E6 induces a Warburg effect by interrupting the association between HIF1α and the tumor suppressor von Hippel–Lindau (VHL) and E3 ligase, thus stabilizing HIF1α [201], and has also been reported to interact with cMyc [202,203]. Altered PI3K/Akt signaling was further associated with mitochondrial uncoupling and induction of oxidative stress [204]. In addition to E6 and E7, E5 was also shown to induce glycolysis by stimulating ERK1/2 and Akt signaling [205]. The E2 proteins, negative regulators of E6 and E7, localize to mitochondria, where they induce oxidative stress and modulate OXPHOS [206]. HPV18 E2 was found to localize to several respiratory complexes in mitochondria and this was associated with mitochondrial damage and increased ROS production, which stabilized HIF1α and induced glycolysis [207]. Analysis of the metabolic secretome of one normal and three cancerous cervical cell lines, of which two were HPV positive, showed that the HPV+ cancer cell lines exhibited features of Warburg metabolism, with increased glucose metabolism and upregulated PPP activity. Furthermore, HPV+ cancer cell lines displayed a profile of cysteine/glutathione metabolites that suggested that they differentially deploy glutathione metabolism to maintain a favorable cellular redox balance [208]. Analysis of TCGA data disclosed upregulation of several genes involved in FA metabolism in HPV-positive head and neck squamous cell carcinoma in comparison to HPV-negative cases [209].

### 6.3. Human T Cell Leukemia Virus 1 (HTLV-1)

HTLV-1, the causative agent of adult T cell leukemia/lymphoma (ATL), is the first retroviral agent known to induce human cancer [210]. In addition to classical retroviral genes, the viral genome encodes two oncoproteins, the transactivator protein Tax and the helix basic zipper proteins, HBZ [211]. After infection and viral entry, reverse transcription of the single-stranded RNA genome takes place in the cytoplasm and the linear double-stranded viral DNA is integrated into the host cell genome. The integrated proviral DNA can be latent or it may be actively transcribed [212]. The virus uses GLUT1 as the entry receptor without clear metabolic alteration after infection [213]. However, hypoxia, frequently encountered by circulating T cells in the lymphoid organs and bone marrow, significantly enhances HTLV-1 reactivation from latency. Furthermore, it has been shown that culturing naturally infected CD4+ T cells in glucose-free medium or inhibition of glycolysis or the mitochondrial electron transport chain strongly suppress HTLV-1 transcription, suggesting that these metabolic processes regulate HTLV-1 proviral latency and reactivation in vivo [214].

Tax and Hbz are two viral oncogenes known to be essential for cellular transformation. Tax is known to activate the PI3K/Akt and NF-κB pathways and inhibits the tumor suppressor p53 [215]. Tax expression is important for the lymphoproliferative and immortalizing effect of HTLV-1 infection [216,217]. Tax increases ROS production, which induces genomic DNA damage and cellular senescence. Additionally, the Tax-dependent activation of NF-κB signaling induces cellular senescence. This effect can be countered through co-expression of HBZ [218]. Both Tax and HBZ induce aberrant cell proliferation and apoptosis [219]. HBZ was shown to activate the mTOR signaling pathway [220]. Recently, the HTLV-1 p30^II^ protein was found to suppress Tax- and HBZ-induced oxidative stress and mitochondrial damage [220]. However, a recent study showed that GR expression and activity and glutathione levels in HTLV-1 patients were reduced compared to healthy control patients, and this reduction was negatively correlated to viral loads [221]. These findings confirm the role of HTLV-1-induced oxidative stress and mitochondrial damage in viral pathogenesis.

Finally, the Hippo signal effector, YAP, known for its implication in the regulation of glutaminolysis, was found to be activated in ATL and its activation to be essential to maintain cellular proliferation of HTLV-1-infected cells. The activation was found to be induced via Tax protein through NF-κB [222].

### 6.4. Hepatitis B Virus (HBV)

HBV, a hepatotropic, enveloped DNA virus from the family *Hepadnaviidae*, is a major viral cause of hepatocellular carcinoma [223]. About 300 million people are chronically infected with HBV worldwide. The viral genome harbors four overlapping open reading frames (ORFs) named C, P, S, and X. The ORF C encodes the core protein (HBc) and its related proteins E antigen (HBe) and the precure protein (p22cr), the ORF P encodes the polymerase (pol), the ORF S encodes three types of surface antigens (HBs), and lastly, the ORF X encodes the oncogenic protein X (HBx) [224,225]. After HBV infection, HBV DNA is converted into covalently closed circular DNA (cccDNA), which accumulates in the nucleus as a stable episome, leading to persistent infection [226]. Both immunological and viral factors participate in the development of HBV-induced hepatocellular carcinoma. Chronic infection induces oxidative stress and inflammation that trigger liver fibrosis, characterized by the accumulation of extracellular matrix proteins, including collagen. In the long term, this leads to degradation of the liver microenvironment and cancer development [227]. However, integration of the viral DNA into the host genome can lead to HCC independently of liver damage [228].

The oncogenic viral protein HBx in addition to the pre-S and S polypeptides can activate the PI3/Akt/mTOR, Jak/Stat, Pyk2, Wnt/β-catenin, and the EGFR/RAS/MAPK pathways [229]. HBx also inhibits the tumor suppressors p53 and RB. HBx transgenic mice showed impaired glucose metabolism and increased expression of genes involved in gluconeogenesis [230]. However, in an in vitro model for HBV replication, increased expression of glycolytic factors (FBP aldolase, TPI, PGK1, G6P isomerase) and TCA cycle enzymes (MDH, CS, SDH) and elevation of key glycolytic and TCA metabolites has been reported [231,232]. Increased glycolysis resulted in the activation of nucleotide synthesis [231]. Upregulation of G6PD, the first and rate-limiting enzyme of the PPP, by HBx has been reported [231,233].

Increased glucose uptake, glycolysis, and lactate production were also reported to be induced by a pre-S2 mutant in an mTOR-dependent fashion [234,235]. Using multi-omics analyses to characterize the HBc-transfected hepatoma cell line HepG2, HBc protein was shown to enhance amino acid, lipid, and glucose metabolism in hepatocellular carcinoma [236]. HBc was shown to bind directly and activate enzymes involved in glycine metabolic pathways and phenylalanine degradation. The ability of HBc to modulate glucose and lipid metabolism was found to be regulated via Max-like protein (MLX), which can be recruited to the nucleus in an HBc-dependent manner. MLX binds to and upregulates the glycolytic proteins aldolase C and phosphoenolpyruvate carboxykinase [236]. Knocking down MLX was shown to inhibit Myc-induced metabolic changes and apoptosis, suggesting a role of MLX protein in glycolysis and lipid metabolism in carcinogenesis [237].

In respect to amino acid metabolism, HBV infection was found to significantly increase the expression of glutamine synthetase in liver tissues compared to normal control livers from uninfected patients. Furthermore, in HBV patients with HCC, glutamine synthetase levels were higher than in HBV carriers without HCC [238]. HBx protein has been shown to induce mitochondrial fragmentation via Drp1 stimulation, and mitophagy via parkin, PINK1, and LC3B stimulation [239]. Lipid accumulation, induced by HBV or due to mitochondrial dysfunction, is associated with an increase in HBV gene expression. This process implies induction of lipogenic transcription factors such as sterol regulatory element-binding protein 1 (SREBP-1) and peroxisome proliferator-activated receptor γ [240]. HBV transgenic mice showed increased expression of factors involved in lipid and steroid biosynthesis and metabolism, including retinol-binding protein 1, SREBP2, ATP citrate lyase, and FASN [241,242]. Furthermore, HBV replication and release are sensitive to FASN inhibitors [243,244]. However, HBx has also been reported to stimulate FA oxidation [245]. Finally, HBx overexpression was shown to induce lipid accumulation in HepG2 cells and HBx transgenic mice by the induction of lipogenic transcription factors and FA-binding protein 1 [246,247,248,249] and inhibition of lipid secretion [250], suggesting that it can induce hepatic steatosis. Furthermore, HBV-replicating cells showed elevated levels of enzymes involved in phosphocholine synthesis [231]. In vitro and in HBV-infected humanized mice, elevated levels of factors related to uptake (LDLR), biosynthesis (HMGCR) and transcriptional regulation (SREBP2) of cholesterol [251,252] and hCYP7A1, the rate-limiting enzyme for the conversion of cholesterol into bile acids [251], were detected.

### 6.5. Hepatitis C Virus (HCV)

HCV, a hepatotropic single-stranded positive-sense RNA virus from the family *Flaviviridae*, is another important cause of hepatocellular carcinoma [223]. About 1% of the world population are chronic HCV carriers. The viral genome consists of a single large ORF that encodes a single polyprotein that is processed by host and viral proteases to yield the structural proteins core and two envelope glycoproteins as well as seven non-structural proteins (NS) [253].

Similar to HBV infection, chronic HCV infection is associated with oxidative stress and inflammation, which triggers fibrosis and leads to carcinogenesis on the long term [227]. Moreover, chronic HCV infection is generally associated with metabolic disorders such as insulin resistance and steatosis [254,255]. HCV-induced steatosis is associated with increased risk of hepatocellular carcinoma, particularly in the context of genotype 3 [256].

HCV infection was found to activate the N-Ras/PI3K/Akt/mTOR pathway [257] and to inhibit the tumor suppressor p53 [258]. HCV core, NS3, NS5A, and NS5B proteins are responsible for activating PI3/Akt/mTOR and the EGFR/RAS/MAPK pathways and inhibiting the pro-apoptotic proteins p53 and RB [229]. Furthermore, HCV infection in vitro is characterized by increased glucose consumption and lactate production, confirming the activation of glycolytic fluxes, thought to be due to stabilization of HIF1α [259,260,261]. A proteome analysis using an HCV cell culture model showed upregulation of key enzymes involved in glycolysis, the PPP, and the TCA cycle, favoring biosynthetic pathways [262]. NS5A interacts with and activates HK2 in the hepatoma cell line Huh7.5, increasing glycolysis [259]. Additionally, the D2 domain of NS5A is able to activate glucokinase isoenzyme, thus inducing lipogenesis in liver cells. Glucokinase is the predominant isoenzyme of hexokinase in non-cancer cells [263]. Recent studies showed the importance of an active glycolytic pathway for viral replication and the release of virions from infected cells. Dicholoroacetate, a PDK inhibitor that allows reshuttling of pyruvate towards the TCA cycle, suppressed viral replication [264]. Huh7 cells cultured in galactose instead of glucose did not support viral release due to inhibited glycolysis and activation of OXPHOS [265]. However, contradictory data show that HCV reduces GLUT surface expression and induces gluconeogenesis [266].

HCV NS5A was found to affect lipid metabolism by targeting the AMPK/SREBP-1 pathway [267]. Activation of SREBPs was shown to be due to PI3K/Akt signaling [268]. Interestingly, HCV replication is modulated by the interaction between NS5B and FASN, confirming the importance for lipid metabolic reprogramming for HCV persistence [269,270,271]. Furthermore, additional factors and enzymes involved in cholesterol and lipid synthesis, saturation, and secretion have been shown to be essential for HCV replication, virion production, and secretion [272,273,274,275,276,277,278]. Additionally, HCV and, in particular, NS5A promote GLS expression, glutamine uptake, and glutaminolysis in a cMyc-dependent manner [261,279].

Using a combination of metabolic, proteomic, and transcriptomic analyses of HCV-infected cells, increased glucose consumption and metabolism were confirmed. Furthermore, an intracellular accumulation of very-long-chain FAs was detected due to reduced peroxisomal activity [280].

### 6.6. Kaposi Sarcoma-Associated Herpesvirus (KSHV)

KSHV is a large enveloped double-stranded herpesvirus-8, mainly associated with the development of Kaposi’s sarcoma in untreated HIV patients and other immunocompromised patients [281]. The viral genome harbors about 100 genes [282]. As a herpesvirus, KSHV is capable of latency and lytic replication. In latent infection, the viral genome persists as a multicopy extrachromosomal circular episome. Cancer cells are usually latently infected and express viral oncogenes such as latency-associated nuclear antigen (LANA), viral cyclin (v-cyclin), and Kaposin [283].

KSHV LANA is known to inhibit the tumor suppressors p53 and RB [284,285]. Microvascular endothelial (TIME) cells infected with KSHV show enhanced glucose uptake, glycolysis, and lactate production and reduced oxygen consumption rates, suggesting lower OXPHOS activity [286]. Inhibiting LDH activity with oxamate showed a reduction in KSHV viral production after lytic infection in TIME cells, which indicates the importance of glycolysis for KSHV viral particle production [287]. Furthermore, oxamate reduced the expression of early and late viral genes such as open reading frame (ORF) 45 and ORF 59 (early genes) and ORF26 and K8.1 (late genes) [287]. KSHV infection has been shown to induce glycolytic metabolism by several groups via increased HIF1α expression and stabilization [288,289,290]. The virus-encoded G-protein-coupled receptor (vGPCR) is a direct target of HIF1α and is a major inducer of metabolic changes induced by KSHV infection. Viral mutants lacking vGPCR are unable to induce metabolic changes [291]. PI3K/Akt/mTOR is another signaling pathway stimulated by KSHV that leads to increased glycolysis [292]. Inhibition of PI3K/Akt was found to decrease the glycolytic fluxes in infected cells [293]. KSHV also induced the expression of PPP enzymes, including G6PD, TKT, and TALDO, in a nuclear elongation factor 2 (Nrf2)-dependent fashion. Nrf2 was not only found to be induced in infected cells but also in Kaposi’s sarcoma infection-associated lesions [294].

Mass spectrometric analysis of KSHV-infected TIME cells showed an increase in long-chain FA production. While the precursors of FA synthesis, choline and phosphocholine, were also increased, there was a decrease in the degradation products of phospholipids, glycerophosphorylcholine, and glycerol-3-phosphate. Exposure of infected TIME cells to FASN inhibitors significantly increased cell death [295]. However, FA synthesis in Kaposi’s sarcoma is downregulated and not needed for the expression of early and late KSHV genes [287]. Overproduction of cholesterol esters is shown in latent KSHV infections and inhibition of cholesteryl esterification impairs neo-angiogenesis, suggesting a role of the cholesterol ester biosynthetic pathway in Kaposi’s sarcoma development [296]. Nevertheless, RNA sequencing analysis of Kaposi’s sarcoma biopsies showed a significant downregulation of several lipid metabolic pathways, including FASN, compared to normal tissues [297].

Cells latently infected with KSHV are glutamine addicted and depend on glutaminolysis for survival. Lytic infection also depends on glutaminolysis [287,298]. KSHV infection increases glutamine uptake by upregulating the expression of glutamine receptor SLC1A5. Furthermore, the enhanced glutamine metabolism is due to overexpression of GLS, GDH1, and glutamic-oxalacetate transaminase 2 enzymes, essential for glutaminolysis [299]. Instead of fueling the TCA cycle, glutaminolysis in KSHV-infected cells serves as nitrogen supply for nucleic acid biosynthesis [299]. Moreover, metabolic analysis on three-dimensional cell cultures showed that KSHV infection enhanced nonessential amino acid and in particular proline and the associated polyamine metabolism [300]. Hypusine, a polyamine-derived amino acid, was shown to enhance LANA expression via hypusination of the eukaryotic initiation factor 5 A (eIF5A) [301,302]. Moreover, inhibition of polyamine biosynthesis or eIF5A hypusination decreased LANA expression, reduced viral episomal maintenance, and suppressed viral infection [301,302].

### 6.7. Merkel Cell Polyomavirus (MCV)

MCV is a non-enveloped, double stranded DNA virus that is thought to be the cause of the majority of Merkel cell carcinoma cases. The viral genome harbors early and late gene regions on opposite strands, with a central non-coding control region containing the origin of replication [303]. After viral infection, the early region is expressed immediately and undergoes splicing into two T antigen oncoproteins, large and small. The late region encodes the viral structural capsid proteins, viral protein 1 (VP1) and viral protein 2 (VP2) [304]. MCV infection is usually persistent and asymptomatic. Nevertheless, VP1 antibodies can be detected in healthy subjects and Merkel cell carcinoma patients but at lower levels [305]. Antibodies against T antigens are hardly detected except in Merkel cell carcinoma patients and can be used to monitor disease progression [306].

Viral T antigens are expressed persistently in Merkel cell carcinoma and participate in the process of tumorigenesis. T antigens were found to inhibit the tumor suppressors RB and p53 and stabilize the cMyc signaling pathway [307,308,309]. Transcriptome analysis of normal human fibroblasts with inducible expression of small T antigen showed upregulation of glycolytic genes and the monocarboxylate lactate transporter SLC16A1. These cells showed increased lactate production, suggesting a reduced OXPHOS activity [310].

## 7. Conclusions

This review outlines the similarities of the metabolic reprogramming observed in transformed cells and cells infected with oncogenic viruses. Indeed, viral infection and amplification require the use of the cellular metabolic machinery to synthesize viral proteins, nucleic acids, and lipids, similar to the biosynthetic needs for cell proliferation. Furthermore, redox homeostasis in infected cells needs to be controlled in order to balance its effects on viral replication versus host cell survival and thus viral persistence. Similarly, cancer cells are known to depend on a tight regulation of oxidative stress. Finally, the secretion of certain metabolites (here, we discussed only lactate) is important to ensure cell survival for both cancer and infected cells in an inflammatory microenvironment.

Metabolic reprogramming is not exclusive to oncogenic viruses. Similar changes are often observed in the context of infections with non-oncogenic viruses. The oncovirus-induced metabolic changes outlined here are therefore not sufficient for the induction of carcinogenesis. However, in the context of inflammation and oxidative stress, the presence of “pro-cancerogenic” metabolism is likely to facilitate malignant transformation induced by viruses.

Specific metabolic dependencies in cancer have been the basis for effective therapeutics, including inhibitors that target IDH, and folate and thymidine metabolism [311]. Many of these inhibitors have been tested for their efficacies to inhibit viral replication, as discussed above and detailed in Table 1, suggesting that targeting the host cell metabolism is a therapeutic strategy for viral infections. Furthermore, in some cases, dual therapeutic effects in the context of oncogenic infections have been observed: the inhibition of viral replication and the spread and elimination of transformed cells. These findings warrant further exploration of the dependencies of viruses on host cell metabolism, particularly for viruses where no direct-acting antiviral that specifically targets the virus exists.

## Figures and Tables

**Figure 1 cancers-14-05742-f001:**
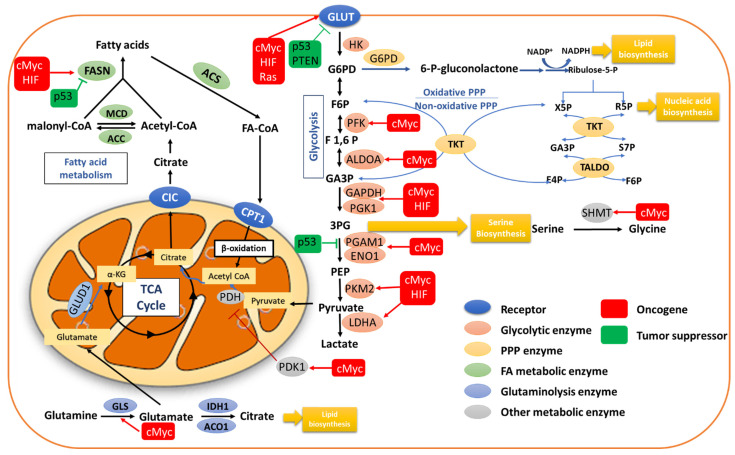
**Reprogramming of cell metabolism in cancer**. Metabolic pathways regulating glucose metabolism, including glycolysis, TCA cycle and PPP, fatty acid metabolism, and glutaminolysis, are generally altered in malignant transformation by altered signaling pathways, oncogenes, and tumor suppressor genes. Abbreviations: GLUT, glucose transporter; HK, hexokinase; G6P, glucose 6-phosphate; F6P, fructose 6-phosphate; F1,6P, fructose 1,6-biphosphate; ALDOA, aldolase A; GA3P, glyceraldehyde 3-phosphate; GAPDH, glyceraldehyde 3-phosphate dehydrogenase; PGK1, phosphoglycerokinase 1; PGAM1, phosphoglycerate mutase 1; ENO1, α-enolase; PEP, phosphoenol pyruvate; PKM2, pyruvate kinase isozyme type 2; LDHA, lactate dehydrogenase A; PKD1, pyruvate dehydrogenase kinase 1; PDH, pyruvate dehydrogenase; PPP, pentose phosphate pathway; G6PD, glucose 6-phosphate dehydrogenase; NADPH, nicotinamide adenine dinucleotide phosphate; X5P, xylulose 5-phosphate; R5P, ribose 5-phosphate; S7P, sedoheptulose 7-phosphate; E4P, erythrose 4-phosphate; TKT, transketolase; TALDO, transaldolase; CIC, citrate carrier; CPTI, carnitine palmitoyl transferase 1; MCD, malonyl-coA decarboxylase; ACC, acetyl-coA carboxylase; FASN, fatty acid synthase; ACS, acetyl-coA synthetase; α-KG, α-ketoglutarate; GLS, glutaminase; GLUD1, glutamate dehydrogenase 1; IDH1, isocitrate dehydrogenase 1; Aco1, aconitase 1; 3PG, 3-phosphoglycerate; SHMT, serine hydroxymethyltranferase.

**Figure 2 cancers-14-05742-f002:**
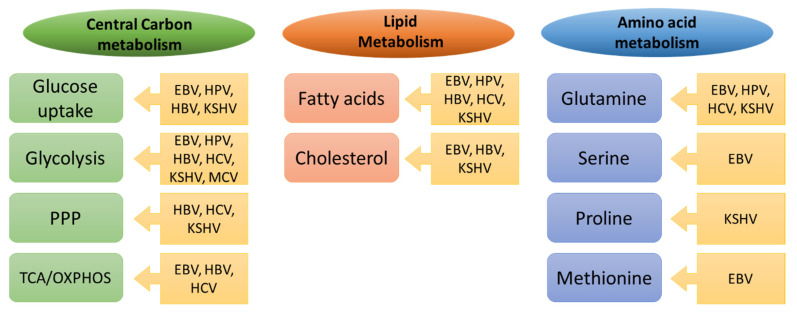
**Common metabolic alterations in cancer and oncoviral infections.** The key metabolic pathways altered in tumors and targeted by the indicated oncoviruses are depicted. See text for details.

**Figure 3 cancers-14-05742-f003:**
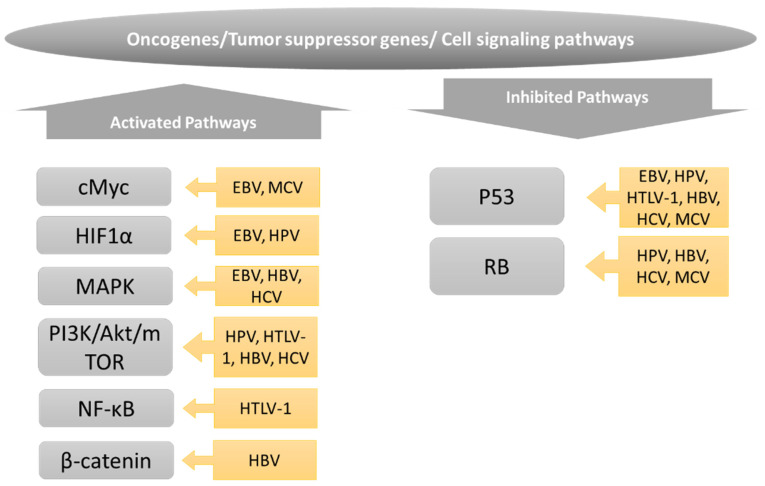
**Oncogenes, tumor suppressors, and transcription factors implied in the metabolic alterations associated with oncogenic transformation and oncoviral infections**. See text for details.

**Table 1 cancers-14-05742-t001:** **Antiviral effects of metabolic inhibitors**.

Metabolic Drug	Target Enzyme	Virus	Model/Cell System	Outcome	Reference
STF-31	GLUT1	EBV		kills NPC cells in vitro	[175]
2-deoxy-D-Glucose	Hexokinase, PI3K/Akt	HTLV-1	PBMCs	suppresses viral transcription	[214]
		HBV	Huh7	suppresses viral protein production (HBV delivered with adenovirus)	[312]
oxamate	LDHA	KSHV	TIME	decreases virion production	[287]
dichloroacetate	PDK	HCV	Huh7.5	decreases viral replication	[264]
rapamycine	mTOR	EBV	cell type dependent	induces of lytic cycle	[181]
CB-839/BPTES	Glutaminase	HCV	Huh7.5	inhibits replication and infection establishment	[261]
		KSHV	TIME	decreases virion production	[287]
difluoromethylornithine	ornithine decarboxylase 1	KSHV	TIME/BCBL-1	decreases virion production	[302]
N1-guanyl-1,7-diaminoheptane (GC7)	deoxyhypusine synthase	KSHV	TIME/BCBL-1	decreases virion production	[302]
cerulenin	FASN	HCV	Huh7	decreases viral replication	[313]
C75	FASN	HCV	Huh7	decreases virion production	[270]
GSK1995010	FASN	HBV	HepG2.2.15.7	decreases viral replication	[243]
4-(1-(5-(2-cyclopropyl-4-methyl-1H-imidazol-5-yl)- 2,4-dimethylbenzoyl)-3-fluoroazetidin-3- yl)benzonitrile	FASN	HBV	HepG2^NTCP^	inhibits HBs secretion	[244]
LCQ908/pradigastat	diacylglycerol O acyltransferase 1	HCV	clinical phase trial/Huh7.5	inhibits replication in vitro but not in vivo	[274]
BMS-200150/BMS-20101038	microsomal transfer protein	HCV	Huh7.5	decreases virion release	[275,276]
3J [314]	Stearoyl-CoA desaturase	HCV	Huh7.5	decreases viral replication and assembly	[277]
25-hydroxycholesterol	SREBP	HCV	Huh7	decreases viral replication	[313]
5-tetradecyloxy-2-furoic acid (TOFA)	acetyl-coA carboxylase	KSHV	TIME	decreases virion production	[287]
CP640186	acetyl-coA carboxylase	HBV	HepG2.2.15.7	decreases viral replication	[243]
2-(1-((R)-2-(((1s,4S)-4-hydroxycyclohexyl)oxy)-2- (2-methoxyphenyl)ethyl)-5-methyl-6-(oxazol-2-yl)- 2,4-dioxo-1,4-dihydrothieno[2,3-d]pyrimidin-3(2H)- yl)-2-methylpropanoic acid	acetyl-coA carboxylase	HBV	HepG2^NTCP^	inhibits HBs secretion	[244]
(R)-4-((diethylamino)methyl)-N-(2- methoxyphenethyl)-N-(pyrrolidin-3-yl)benzamide	subtilisin kexin isozyme-1/site-1 protease	HBV	HepG2^NTCP^	inhibits HBs secretion	[244]

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
