# Peer review of "An Update on the Metabolic Landscape of Oncogenic Viruses"

_cancers, 2022, doi:10.3390/cancers14235742_

Round 1

Reviewer 1 Report

This is a well written review about the role of metabolic pathways altered in tumors caused by oncogenic viruses. The first part exhaustively reviews the biochemical pathways and then talks about the virus/viral proteins that mediate these effects. The idea of metabolic deregulation in cancer cells has been known for a long time. However, there are fewer studies about this reprogramming of cells in tumors which are primarily virus driven. This review tries to compile the existing data on how virus infected cells/viral oncoproteins target the cell metabolism to drive the tumor proliferation and survival. I have only some minor comments  

Comments: The section Role of oncogenes, tumor suppressors and cell signaling in regulating cancer cell metabolism seems to be not relevant to oncoviruses. The authors can either incorporate this information in the respective virus sections if these are deregulated.

A table of currently used inhibitors / in clinical trial/ cell culture/pre-clinical model against these viruses could be made!

Line 350: E6 degrades P53 while E7 degrades Rb. Rephrase the sentence to correct it

HPV E2 causes metabolic reprogramming. cite relevant literature.

Author Response

We would like to thank the Reviewers for their time and effort to read and review manuscript and for their constructive comments. We have tried our best to address all issues in the revised manuscript as follows:

Reviewer 1

This is a well written review about the role of metabolic pathways altered in tumors caused by oncogenic viruses. The first part exhaustively reviews the biochemical pathways and then talks about the virus/viral proteins that mediate these effects. The idea of metabolic deregulation in cancer cells has been known for a long time. However, there are fewer studies about this reprogramming of cells in tumors which are primarily virus driven. This review tries to compile the existing data on how virus infected cells/viral oncoproteins target the cell metabolism to drive the tumor proliferation and survival. I have only some minor comments  

Comments: The section Role of oncogenes, tumor suppressors and cell signaling in regulating cancer cell metabolism seems to be not relevant to oncoviruses. The authors can either incorporate this information in the respective virus sections if these are deregulated.
We agree with the Reviewer that this section seemed to be somewhat unconnected. We have added a short introduction to this section to better connect it to the whole text (page 7).

A table of currently used inhibitors / in clinical trial/ cell culture/pre-clinical model against these viruses could be made!
We thank the Reviewer for this proposition. A table listing the antiviral effects of metabolic inhibitors with references is now included in the manuscript.

Line 350: E6 degrades P53 while E7 degrades Rb. Rephrase the sentence to correct it
We corrected the sentence (Page 10 line 422). The sentence now reads: “Moreover, E7 degrades the tumor suppressor RB [195] and both E6 and E7 inhibit p53 [196-198] and activate the PI3K/Akt/mTOR signaling pathway [199].”

HPV E2 causes metabolic reprogramming. cite relevant literature.
We thank the reviewer for flagging out this omission.  We have added the following information (including new references) to the HPV section: “Besides E6 and E7, E5 was shown also to induce glycolysis by stimulating ERK1/2 and Akt signaling [204]. The E2 proteins, negative regulators of E6 and E7, localize to mitochondria, where they induce oxidative stress and modulate oxidative phosphorylation  [205].“

Reviewer 2 Report

In this review, authors discuss that oncogenic virus infection and cancer development share many resemblances regarding metabolic reprogramming. Authors provide in-depth summary of multiple similarities emerged between metabolic reprogramming by oncogenic viruses and cancers. The viruses discussed include Epstein-Barr virus (EBV), Human papilloma virus (HPVs), Human T cell lymphotropic virus-1 (HTLV-1), Hepatitis B and C viruses (HBV and HCV, respectively), Kaposi’s sarcoma herpesvirus (KSHV), and Merkel Cell Polyomavirus (MCV). This reviewer believes that this paper is defining how oncogenic virus infection alters host metabolism and will contribute to better understandings in virology, viral pathogenesis, and cancer development. It may contribute to bring insight to develop the treatment discoveries for these oncogenic viruses. The manuscript is well-written but has some issues should be addressed.

Major comments

1.     As the paper title is saying, authors should provide more summary of very recent updated findings regarding the role of host metabolism during oncogenesis of human viruses. There are relevant recent works that are not cited. For example...

-        EBV and methionine : Guo et al., 2022, Cell Metabolism 34, 1280–1297

-        EBV and Cytidine : Liang et al., 2021, mBio, Vol. 12, No. 4

-        Metabolic rewiring by HPV : Pappa et al., Scientific Reports volume 11, Article number: 17718 (2021)

-        KSHV and polyamine : Choi et al., Cell Reports Volume 40, Issue 7 / Fiches et al., Plos pathogens https://doi.org/10.1371/journal.ppat.1010503

-        KSHV K1 and proline : Choi et al., PNAS 117 (14) 8083-8093

2.     It would be helpful to add one figure showing the reprogramming of cell metabolism by oncogenic viruses to compare similarities of metabolic reprogramming by oncogenic viruses and cancers.

Minor comments

1.     Line 71 : “Kaposi's sarcoma herpesvirus (KSV)” (KSHV)

2.     Line 74 : “During these prolonged infections, oncogenic viruses manipulate cell signaling pathways that control cell cycle progression, apoptosis and inflammation.” Exact same sentence can be found in abstract, too.

3.     Line 126 : “Switching the expression from PKM1 to PKM2 reverses the Warburg effect,” cited paper is saying PKM2 to PKM1 switching.  

4.     Line 211 : In the section of Reprogramming of amino acid metabolism, they make the claim that several amino acid were shown to have importance in cancer metabolism, however authors only describe glutamine. They should discuss more.

5.     Manuscript would benefit from additional editing for more consistent language usage (e.g. GLUT1 and GLUT-1 used).

Author Response

Reviewer 2

In this review, authors discuss that oncogenic virus infection and cancer development share many resemblances regarding metabolic reprogramming. Authors provide in-depth summary of multiple similarities emerged between metabolic reprogramming by oncogenic viruses and cancers. The viruses discussed include Epstein-Barr virus (EBV), Human papilloma virus (HPVs), Human T cell lymphotropic virus-1 (HTLV-1), Hepatitis B and C viruses (HBV and HCV, respectively), Kaposi’s sarcoma herpesvirus (KSHV), and Merkel Cell Polyomavirus (MCV). This reviewer believes that this paper is defining how oncogenic virus infection alters host metabolism and will contribute to better understandings in virology, viral pathogenesis, and cancer development. It may contribute to bring insight to develop the treatment discoveries for these oncogenic viruses. The manuscript is well-written but has some issues should be addressed.

Major comments

As the paper title is saying, authors should provide more summary of very recent updated findings regarding the role of host metabolism during oncogenesis of human viruses. There are relevant recent works that are not cited. For example...

We thank the Reviewer for pointing us to this literature. The corresponding sections have been updated as follows:

-        EBV and methionine: Guo et al., 2022, Cell Metabolism 34, 1280–1297

“However, methionine restriction, or methionine cycle perturbation, were shown to cause hypomethylation of EBV genomes and to de-repress latent membrane protein and lytic gene expression in Burkitt cells. Additionally, methionine metabolism was shown to regulate EBV latency genes required for B cell immortalization [179=Guo et al. 2022].”

-        EBV and Cytidine: Liang et al., 2021, mBio, Vol. 12, No. 4

 “In respect to the nucleotide metabolism, EBV infection of primary B cells was found to upregulate cytidine triphosphate synthases CTPS1 and CTPS2, which is consistent with the observation that purine dNTP biosynthesis is critical in the early stages of EBV-mediated B-cell immortalization [181=Hafez et al. 2017]. Furthermore, EBNA2 and/or EBNA-LP were needed for the induction of CTPS1 in newly infected B cells. Finally, CTPS1 depletion impaired EBV lytic DNA synthesis [182 = Liang et al 2021].”

-        Metabolic rewiring by HPV: Pappa et al., Scientific Reports volume 11, Article number: 17718 (2021)

“Analysis of the metabolic secretome of one normal and three cancerous cervical cell lines, of which two were HPV positive, showed that the  HPV+ cancer cell lines exhibited features of Warburg metabolism with increased glucose metabolism and upregulated PPP activity. HPV+ cancer cell lines furthermore displayed a profile of cysteine/glutathione metabolites that suggested that they differentially deploy glutathione metabolism to maintain a favorable cellular redox balance [207=Pappa et al 2021].”

-        KSHV and polyamine: Choi et al., Cell Reports Volume 40, Issue 7 / Fiches et al., Plos pathogens https://doi.org/10.1371/journal.ppat.1010503

-        KSHV K1 and proline: Choi et al., PNAS 117 (14) 8083-8093 (lines 579-581)

“Moreover, metabolic analysis on three-dimensional cell cultures showed that KSHV infection enhanced nonessential amino acid and in particular proline and the associated polyamine metabolism [297= Choi et al PNAS 2020]. Hypusine, a polyamine-derived amino acid, was shown to enhance LANA expression via hypusination of the eukaryotic initiation factor 5A (eIF5A) [298=Fiches PloS Path 2022, 299=Choi et al Cell Reports 2022]. Moreover, inhibition of polyamine biosynthesis or eIF5A hypusination decreased LANA expression, reduced viral episomal maintenance and suppressed viral infection [298,299].”

It would be helpful to add one figure showing the reprogramming of cell metabolism by oncogenic viruses to compare similarities of metabolic reprogramming by oncogenic viruses and cancers.

Figures outlining pathways, oncogenes and tumor suppressors targeted by both oncogenic viruses and oncogenic transformation were created and added to the manuscript (see Figures 2 and 3).

Minor comments

  1. Line 71 : “Kaposi's sarcoma herpesvirus (KSV)” à(KSHV)

corrected

  1. Line 74 : “During these prolonged infections, oncogenic viruses manipulate cell signaling pathways that control cell cycle progression, apoptosis and inflammation.” Exact same sentence can be found in abstract, too.

To remove the redundancy, the sentence in the introduction was rephrased: “Although the mechanisms of viral oncogenesis differ, all these oncogenic viruses share the ability to establish persistent chronic infections, often with no obvious symptoms for years, during which they manipulate and alter  cell signaling pathways that coordinate cell cycle progression, apoptosis and inflammation [13].”

  1. Line 126 : “Switching the expression from PKM1 to PKM2 reverses the Warburg effect,” àcited paper is saying PKM2 to PKM1 switching.  

We thank the Reviewer for pointing out this error. We corrected to “switching the expression from PKM2 to PKM1”

  1. Line 211 : In the section of Reprogramming of amino acid metabolism, they make the claim that several amino acid were shown to have importance in cancer metabolism, however authors only describe glutamine. They should discuss more.

Besides glutamine, we now discuss oncovirus-induced changes to serine, proline and polyamine metabolism, in the appropriate sections. We have also updated the introduction where we describe changes to serine, proline and polyamine  metabolism and their roles in the context of cancer.

  1. Manuscript would benefit from additional editing for more consistent language usage (e.g. GLUT1 and GLUT-1 used).

Thank you for this comment, we have carefully revised the manuscript in respect to abbreviations.

Reviewer 3 Report

The work by Gaballah and Bartosch aimed to summarize metabolic alterations promoted by oncoviruses in cancer cells and parallel these alterations with those that occur in cancer. The theme is relevant and the article provides a great guide for studies in this field. However, the way the text is structured makes it very difficult to follow and to be read throughout, specially in the discussion of different viruses in the many subsections, which lack contextual connections with one another.

The inclusion of diagrams and/or tables to summarize the main metabolic alterations promoted by the viruses would facilitate the article readability by providing readers efficient ways to consult review the core concepts of the article while reading it.

Also, core metabolic alterations that are common for the different oncoviruses (e.g.: bias to glycolysis and upregulation of FA metabolism) could be emphasized in the text in sections like the introduction, "Metabolic reprogramming in Oncoviruses" and conclusion, since in these sections authors generally only state that these viruses promote metabolic reprograming of cells without citing which alterations are promoted. This would introduce readers to these metabolic changes and prepare them for the more deepened discussion offered in the specific subsections. 

Minor points:

1) The last paragraph of the introduction section is lacking citations to the relevant literature for the vast majority of phrases. 

2) The in the following phrase "Switching the expression from PKM1 to PKM2 reverses the Warburg effect, decreases tumorigenicity and increases oxygen consumption through shunting glycolysis to OXPHOS [50]." the enzymes isoforms are inverted, since the original article shows that shifting the expression from PKM2 to PKM1 promotes the reversion of the Warbug effect.

3) In the phrase "This reaction is inhibited in cancer cells due to the overexpression of the PDH inhibitor, pyruvate dehydrogenase kinase (PDK), leading to reduced acetyl-coA supply to the TCA cycle [51].", pyruvate dehydrogenase kinase (PDK) should be in plural, since it refers to a family of related enzymes. This facilitate the comprehension of the following phrase, which refers to a specific member of this family (PDK2).

4) In the phrase "Lipid requirement is increased during for synthesis of membranes and signaling molecules [8]." it seems that something is missed between "during" and "for".

Author Response

Reviewer 3

The work by Gaballah and Bartosch aimed to summarize metabolic alterations promoted by oncoviruses in cancer cells and parallel these alterations with those that occur in cancer. The theme is relevant and the article provides a great guide for studies in this field. However, the way the text is structured makes it very difficult to follow and to be read throughout, specially in the discussion of different viruses in the many subsections, which lack contextual connections with one another.

The inclusion of diagrams and/or tables to summarize the main metabolic alterations promoted by the viruses would facilitate the article readability by providing readers efficient ways to consult review the core concepts of the article while reading it.
We thank the Reviewer for this comment. We have extended Figure 1 (inserted some more details regarding the serine metabolism), and added two additional Figures, which in a very schematic way point out the metabolic pathways and their major regulators that are targeted by oncoviruses as well as oncogenic transformation. We hope that they will help to give an overview to future readers.

Also, core metabolic alterations that are common for the different oncoviruses (e.g.: bias to glycolysis and upregulation of FA metabolism) could be emphasized in the text in sections like the introduction, "Metabolic reprogramming in Oncoviruses" and conclusion, since in these sections authors generally only state that these viruses promote metabolic reprograming of cells without citing which alterations are promoted. This would introduce readers to these metabolic changes and prepare them for the more deepened discussion offered in the specific subsections.
We have adapted that abstract accordingly, where we now point out: “Here, we draw a parallel between metabolic changes observed in cancers or induced by oncoviruses, with focus on pathways involved in the regulation of glucose, lipid and amino acids.

“.

Minor points:

1) The last paragraph of the introduction section is lacking citations to the relevant literature for the vast majority of phrases. 
Two additional references have been added to this section.

2) The in the following phrase "Switching the expression from PKM1 to PKM2 reverses the Warburg effect, decreases tumorigenicity and increases oxygen consumption through shunting glycolysis to OXPHOS [50]." the enzymes isoforms are inverted, since the original article shows that shifting the expression from PKM2 to PKM1 promotes the reversion of the Warbug effect.
We thank the Reviewer for pointing out this error. We corrected to “switching the expression from PKM2 to PKM1”

3) In the phrase "This reaction is inhibited in cancer cells due to the overexpression of the PDH inhibitor, pyruvate dehydrogenase kinase (PDK), leading to reduced acetyl-coA supply to the TCA cycle [51].", pyruvate dehydrogenase kinase (PDK) should be in plural, since it refers to a family of related enzymes. This facilitate the comprehension of the following phrase, which refers to a specific member of this family (PDK2).
The sentence has been changed to “This reaction is inhibited in cancer cells due to the overexpression of pyruvate dehydrogenase kinases (PDK), a family of PDH complex inhibitors, leading to reduced acetyl-coA supply to the TCA cycle [54].”

4) In the phrase "Lipid requirement is increased during for synthesis of membranes and signaling molecules [8]." it seems that something is missed between "during" and "for".
We thank the Reviewer for his attention. The phrase has been corrected to: "Lipid requirement is increased during tumorigenesis for synthesis of membranes and signaling molecules”.

Round 2

Reviewer 2 Report

The authors have addressed my concerns.

Reviewer 3 Report

The authors have addressed all the raised concerns and the additional figures and table substantially increased the article readability and significance, providing a global integrated views of the article's subjects.